# Diversity Patterns of Wetland Angiosperms in the Qinghai-Tibet Plateau, China

**Yigang Li** [1,2,3], **Yadong Zhou** [4], **Fan Liu** [3], **Xing Liu** [1,2,5,*] **and Qingfeng Wang** [1,2,3,*]

1   College of Science, Tibet University, Lhasa 850000, China
2   Research Center for Ecology and Environment of Qinghai-Tibetan Plateau, Tibet University, Lhasa 850000, China
3   Wuhan Botanical Garden, Chinese Academy of Sciences, Wuhan 430074, China
4   School of Life Sciences, Nanchang University, Nanchang 330000, China
5   College of Life Sciences, Wuhan University, Wuhan 430072, China
*   Correspondence: xingliu@whu.edu.cn (X.L.); qfwang@wbgcas.cn (Q.W.)

**Abstract:** The Qinghai-Tibet Plateau, has a special geological history, diverse habitats, a complex climate, and a large number of wetlands, which harbor a huge of wetland plants. In this study, we sorted out the monographs, literatures and online databases, as well as our own collection from field surveys, and comprehensively combed the checklist and county-level diversity of wetland angiosperms and endangered species in the Qinghai-Tibet Plateau for the first time. The distribution pattern of species richness was analyzed through three groups of environmental variables: energy, water, and habitat. The wetland angiosperms have high richness, with a total of 2329 species, belonging to 91 families and 438 genera, mainly hygrophytes (94.98%). The spatial distribution is uneven, and gradually decreases from the southeast to the northwest of the plateau. Species richness decreased with elevation and latitude and increased with longitude. Annual precipitation (AP) and annual mean temperature (AMT) are the most important variables affecting species diversity. Habitat environmental variables had less influence on species richness distribution and wetland area was not associated with richness distribution. The setting of endangered wetland angiosperm reserves needs to focus on the Hengduan Mountains and southeastern Tibet. Our study provided basic data for the research and protection of wetland plant diversity in the Qinghai-Tibet Plateau.

**Keywords:** wetland angiosperms; diversity; distribution pattern; Qinghai-Tibet Plateau; energy-water

## 1. Introduction

The Qinghai-Tibet Plateau lies in the southwest of China and includes all regions of Tibet Autonomous Region and Qinghai Province, as well as parts of Xinjiang Uygur Autonomous Region, Gansu Province, Sichuan Province, and Yunnan Province [1]. This region is the largest and highest plateau in the world, with an average altitude of more than 4000 m above sea level [2]. Covering an area of about $2.5 \times 10^6$ km$^2$, the Qinghai-Tibet Plateau is surrounded by a series of mountains, such as the Hengduan Mountains, the Himalayas, the Qilian Mountains, and the Karakoram mountains [3,4]. Known as the water tower of Asia, with tens of thousands of glaciers, it is the largest freshwater reserve in the world outside the polar regions [5]. It is also the birthplace of the Yangtze River, the Yellow River, Lancang River, and other important rivers, and plays an important role in the ecological security of the downstream water system [1].

The Qinghai-Tibet Plateau has complex terrain and special natural environment [3]. Affected by the terrain, the climate is complex and changeable, mainly dominated by continental alpine climate [1,6]. Under the influence of Indian and East Asian monsoons, the average temperature and precipitation in the Qinghai-Tibet Plateau show a trend of gradually decreasing from southeast to northwest [4]. The annual precipitation decreases from 2000 mm in the southeast to 50 mm in the northwest [1]. The annual mean temperature is low (−4 °C), which leads to a long time of soil freezing on the surface. Plants only grow

in season for 3 to 4 months in a year [6]. There are obvious temporal and spatial differences in the distribution of atmosphere and heat in the Qinghai-Tibet Plateau. The southeast part is warm and humid, while the northwest part is dry and cold [1]. The main vegetations of Qinghai-Tibet Plateau are meadow and steppe, which account for 24% and 27% of the total area, respectively [7], but other types are also quite abundant across this region, and under the influence of climate and precipitation, from southeast to northwest, the vegetation types gradually changes from forest and shrub to grassland, meadow and desert area [1].

Known as the kidneys of the earth, wetlands are situated between the transitional zones of terrestrial and aquatic ecosystems and carry out a variety of ecosystem functions [8,9]. The wetlands in Qinghai-Tibet Plateau are the largest wetland group in China, covering an area of $13.19 \times 10^4$ km$^2$, accounting for 20% of the total wetland area in China, in which the lake area alone accounts for half of the lake area in China [10] (Figure 1). Alpine wetlands of Qinghai-Tibet Plateau are mainly composed of wet meadow (50%), marshlands (6%), riverine and lacustrine wetlands (44%) [11]. The headwaters regions of the Yangtze River and the Yellow River are the largest frost wetlands in China, and the Zoige wetland is the largest peatland in China. These three wetlands account for more than 85% of the total area of alpine wetlands on the Qinghai Tibet Plateau [12]. In addition, Gao [6] proposed a geomorphic-centered wetland classification system, which divided the Qinghai-Tibet Plateau wetlands into seven types: alpine, piedmont, valley, terrace, floodplain, lacustrine, and riverine.

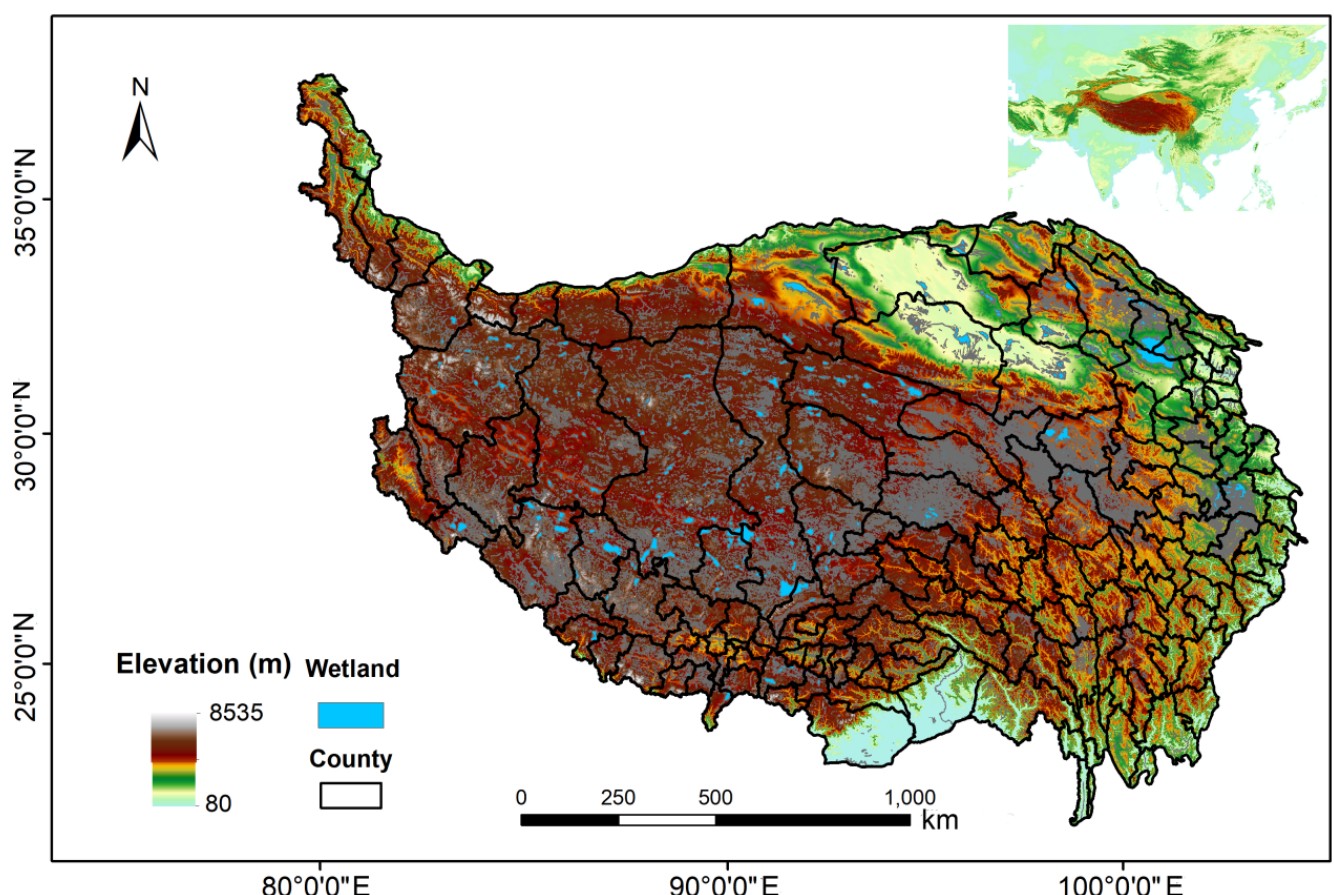

**Figure 1.** Wetland distribution map of Qinghai-Tibet Plateau.

For a long time, the wetland flora of the Qinghai-Tibet Plateau, especially the botanical characteristics and diversity, has attracted extensive attention [13]. Zhao [14] recorded 142 species of swamp plants in Tibet, belonging to 33 families and 84 genera. Cyperaceae, Poaceae, Ranunculaceae, Asteraceae, and Rosaceae accounted for 14%, 10%, 6%, 5%, and 3% of the total

species, respectively. There are about 428 species of seed plants in Qinghai wetland, belonging to 146 genera and 39 families. Poaceae, Ranunculaceae, Cyperaceae, and Asteraceae are the four families with the highest species richness. *Kobresia* and *Carex* are typical representatives of the wetland [15]. Recently, some monographs about the wetland plants in the Qinghai-Tibet Plateau were published successively. *Plateau wetland in Tibet of China* comprehensively introduced the types of wetlands in Tibet, wetland vegetation types, and attached a list of higher plants in Tibet wetland [16]. From 2009 to 2013, China carried out the second national survey of wetland resources, and published a series of monographs, including *China Wetlands Resources_Qinghai Volume* and *China Wetlands Resources_Tibet Volume*. There are 591 species of wetland plants in Tibet Autonomous Region, belonging to 65 families and 205 genera, and 372 species of wetland vascular plants in Qinghai Province, belonging to 46 families and 138 genera [17,18]. Among them, the *China Wetlands Resources_Qinghai Volume* introduces the list and distribution of wetland plants in Qinghai Province, while the *China Wetlands Resources_Tibet Volume* only has the list without distribution information [17,18].

The study of species richness patterns has always been an important topic [19–21]. Scholars have consistently tried to find key variables that affect the distribution and diversity of organisms in different regions, so as to determine the predictors that drive the changes in species richness patterns on different spatial scales [19]. Therefore, people try to explain the mechanism of species distribution with various theories and assumptions [19,22]. The habitat heterogeneity hypothesis believes that heterogeneous habitats can provide more niche space, which provides the possibility for the coexistence of multiple species [23,24]. At the same time, heterogeneous habitats provide shelter for species during climate change, thereby promoting species persistence [23,25]. The energy hypothesis states that regions with higher energy, as measured by annual mean temperature or potential evapotranspiration, have more species [23,26], while the hydrodynamic hypothesis advocates that water and energy work together on species richness [23,27]. In the study of the spatial richness pattern of gymnosperms in China, it was found that energy-water was the best predictor of the richness pattern [19]. A study on the diversity and composition of plant communities in wetlands on the Qinghai-Tibet Plateau found that annual mean temperature and annual precipitation were the main predictors of plant taxonomy [28]. Therefore, we predict that energy and water are the most important factors affecting wetland angiosperms in the Qinghai-Tibet Plateau. Studies have shown that the larger the water surface area, the higher the species richness [29,30]. Larger areas provide more habitat, and over longer periods of time in larger areas, speciation increases and extinctions decrease, so space can affect species richness [31]. When studying the relationship between aquatic plant diversity and the area on the Qinghai-Tibet Plateau, it was found that there is a higher diversity in the middle-range surface area [30]. Therefore, we also speculate that wetland angiosperms diversity in the Qinghai-Tibet Plateau is positively correlated with wetland area.

The Qinghai-Tibet Plateau is an important ecological security barrier in China and even in Asia. The diversity of wetland plants is an important symbol of the structure and function of the wetland ecosystem, and is of great significance to maintaining the stability of the alpine wetland ecosystem on the Qinghai-Tibet Plateau [10,32,33]. Therefore, the study of plant diversity in their wetlands is important for plateau ecosystems. Although there are numerous data on wetland plants studied as described previously, there is currently no comprehensive checklist of wetland angiosperms and information on the distribution pattern of diversity. Based on the field survey, combined with a large number of reference materials, this study comprehensively summarized the checklist and distribution of wetland plants in the Qinghai-Tibet Plateau. Specifically, answer intend to answer the following questions: (1) which species of wetland angiosperms are distributed on the Qinghai-Tibet Plateau? (2) what is the basic information about families and genera? What are the characteristics of plant life forms? (3) The distribution of wetland angiosperms in county-level units on the Qinghai-Tibet Plateau, where are the areas with the highest species richness? (4) What are the endangered wetland angiosperms and where are they mainly

distributed? 5) Which environmental variables are the important driving factors for the diversity distribution of wetland angiosperms?

## 2. Materials and Methods

### 2.1. Construction of Species Checklist and Collection of Distribution Data

We compiled the checklist of wetland angiosperms in the Qinghai-Tibet Plateau and the county-level distribution database through the following steps:

(i) Criteria for evaluating wetland plants: According to the relationship between plants and water, wetland plants are divided into two categories: hydrophytes (high moisture dependence group) and hygrophytes (low moisture dependence group). Hydrophytes were composed of emergent, floating-leaf, floating and submerged plants, which refer to plants living in water, whereas hygrophytes were composed of wet and marshy plants, which are less dependent on water than hydrophytes. They refer to a class of plants that grow in swamps or humid environments. We use the following habitat description keywords as the judgment criteria for hygrophytes: river beach, ditch edge, stream edge, river edge, lake edge, pond, moss bush, tundra, wet grassland, shady and wet place under the forest, shady and wet place in the valley, shady and wet place on the hillside, floodplain, swamp, paddy field, salt lake edge, near the snow line.

(ii) According to monographs, research papers, online databases, and field survey data, the list and distribution map of wetland angiosperms in the Qinghai-Tibet Plateau were compiled. These data sources are as follows: *The Vascular Plants and Their Ecogeographical Distribution of the Qinghai-Tibetan Plateau* [34], *Flora Reipublicae Popularis Sinicae* [35], *Flora of China* [36], *floras of Tibet, Qinghai, Sichuan, Yunnan, Gansu and Xinjiang, China Wetlands Resources (Qinghai Volume, Tibet Volume)* [17,18]. We also supplemented a large number of specimen data and online data, such as National Specimen Information Infrastructure (http://www.nsii.org.cn, accessed on 27 January 2022), Global Biodiversity Information Facility (https://www.gbif.org/, accessed on 22 February 2022), Chinese Virtual Herbarium (https://www.cvh.ac.cn/, accessed on 25 February 2022), Plant Photo Bank of China (http://ppbc.iplant.cn, accessed on 1 May 2021). These monographs and specimen data are a large number of investigations and arrangements made by many scientific researchers on the plants and their distribution on the Qinghai-Tibet Plateau in the past 60 years [37]. We used the following database to calibrate the directory to ensure the accuracy of the directory: China Wetland Plant Database (http://zgsdzw.com, accessed on 1 May 2021), China Aquatic Plant Database (http://www.plant.csdb.cn/aquaticplants, accessed on 14 March 2022). Finally, the field survey data from 2018 to 2020 are supplemented.

(iii) In order to ensure the consistency of the naming of all databases, we used "Species 2000 China Node" (http://www.sp2000.org.cn, accessed on 8 March 2022) [38] and R package "plantlist" to calibrate the checklist of wetland plants on the Qinghai-Tibet Plateau [39]. We recorded the life form, altitude range, habitat information, and county-level distribution points of each species in detail. We used the national Qinghai-Tibet Plateau scientific data center Qinghai-Tibet Plateau urban distribution and urbanization index data set (2018, 2019) to determine the scope of the geographical area of the Qinghai-Tibet Plateau [40].

(iv) According to the Chinese Biodiversity Redlist of Higher Plants, we have compiled the checklist of endangered wetland angiosperms in the Qinghai-Tibet Plateau, and mapped the distribution of endangered wetland angiosperms [41].

### 2.2. Environmental Variables

Based on previous studies, we selected six important climate variables that affect the distribution of species on a large spatial scale [30], which were downloaded from the worldclim database (http://www.worldclim.org/, accessed on 9 June 2022). WA refers to the wetland area of each county-level unit. Ratio_WA is the proportion of the wetland area of county units to the total area of the county, which can reduce the error caused

by the different sizes of the county; The 30 m resolution spatial distribution data set of Chinese marshes and wetlands (2015) is downloaded from the National Earth System Science Data Center (http://www.geodata.cn, accessed on 9 July 2022) [42]. We used ArcGIS to extract water surface data in China from the 1:250000 national basic geographical database, which was downloaded from the National Geomatics Center of China (NGCC, http://www.ngcc.cn/, accessed on 10 July 2022). The elevation variation coefficient (EVC) is the ratio of the standard deviation of the elevation of the county-level unit to the average value, reflecting the relative change of the surface elevation. DEM data from Resource and Environmental Science and Data Center (https://www.resdc.cn/Default.aspx, accessed on 13 June 2022). We divide these variables into three different sets of prediction variables. Among them, energy-related variables include: annual mean temperature (AMT), temperature seasonality (TS), and minimum temperature of the coldest month (MTCM); water-related variables include: annual precipitation (AP), precipitation seasonality (PS), and precipitation of driest month (PDM); Habitat related variables: WA, Ratio_WA, EVC.

*2.3. Statistical Analysis*

We defined species richness (SR) as the number of wetland angiosperms in each county-level geographical unit. In order to eliminate the impact of the area on the species richness of different county-level units, we calculated the species density (SD) of each geographical unit. The calculation equation is as follows:

$$SD = SR/\ln (A) \tag{1}$$

where A is the area of each county-level unit [43].

We used linear regression to analyze the relationship between species diversity patterns and environmental variables We use the "scal" function to standardize environmental variables [44]. The variance inflation factor (VIF) is used to diagnose the collinearity of all environment variables. In the following analysis, all variable's VIF is less than 5 [23]. Due to the strong collinearity between MTCM and other environmental factors, it was not used in the following analysis. We used variance segmentation to analyze the distribution of variability between energy, water, and habitat variables and species richness. The "*vegan package*" is used for variance partitioning analysis [45]. All data analysis is carried out in R 4.2.1 software [46]. The mapping of the spatial pattern of species richness and the extraction of environmental variable data was carried out in ArcGIS 10.2.

## 3. Results

### 3.1. Families, Genera, Species, and Life Forms of Wetland Angiosperms

There are 2329 species of wetland angiosperms in the Qinghai-Tibet Plateau, belonging to 91 families and 438 genera, including 2135 species of herbs and 183 species of shrubs, accounting for 91.67% and 7.86%, respectively; vines and subshrubs account for a very small proportion, with four and seven species, respectively. The top five species-rich families of wetland angiosperms in the Qinghai-Tibet Plateau are Asteraceae, Cyperaceae, Primulaceae, Poaceae, and Gentianaceae, which contain 227, 221, 153, 152, and 105 species, respectively (Figure 2a). The top five species-rich genera are *Carex*, *Primula*, *Pedicularis*, *Impatiens*, and *Corydalis*, which contain 150, 117, 83, 77 and 54 species, respectively (Figure 2b). There are 2212 species of hygrophytes (low moisture dependence), accounting for 94.98% of the total. They are the main types of wetland angiosperms in the Qinghai-Tibet Plateau. There are 117 hydrophytes (high moisture dependence group) in the Qinghai-Tibetan Plateau, which account for a small proportion of wetland angiosperms, and they are mainly distributed in the Hengduan Mountain region, Southeastern Tibetan region, and Northeast of Qinghai Province.

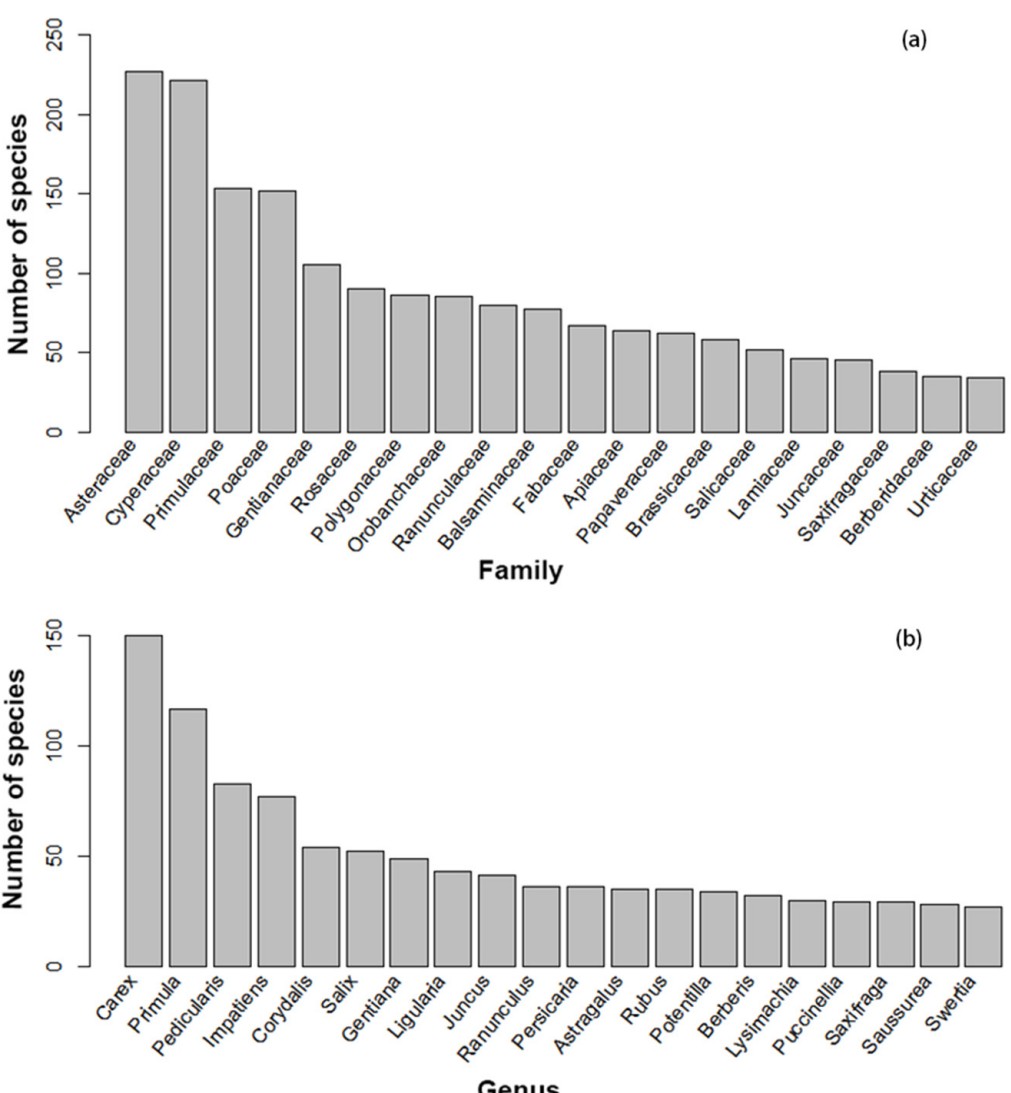

**Figure 2.** The top 20 species-rich families (**a**) and The top 20 species-rich genera (**b**) of wetland angiosperms in the Qinghai-Tibet Plateau.

There are 66 species of endangered wetland angiosperms on the Qinghai-Tibet Plateau, belonging to 27 families and 37 genera, including vulnerable (33), endangered (23), and critically endangered (10). Orchidaceae and Crassulaceae account for a relatively large proportion of endangered plants, with 11 species and 9 species respectively.

### 3.2. County Level Distribution Pattern of Species Richness and Species Density

The spatial patterns of wetland angiosperms species richness (SR) and species density (SD) in the Qinghai-Tibet Plateau were very similar, with obvious regional differences, and showed a downward trend from southeast to northwest. Wetland plant diversity hotspots are mainly concentrated in the Hengduan Mountains in the southeast of the plateau, the Qilian Mountains in the northeast, the Himalayas in the southeast of Tibet, and the San-jiangyuan region of Qinghai. The regions with the lowest diversity are mainly concentrated in the Qaidam Basin, Qiangtang plateau in Tibet, and other regions (Figure 3a,b).

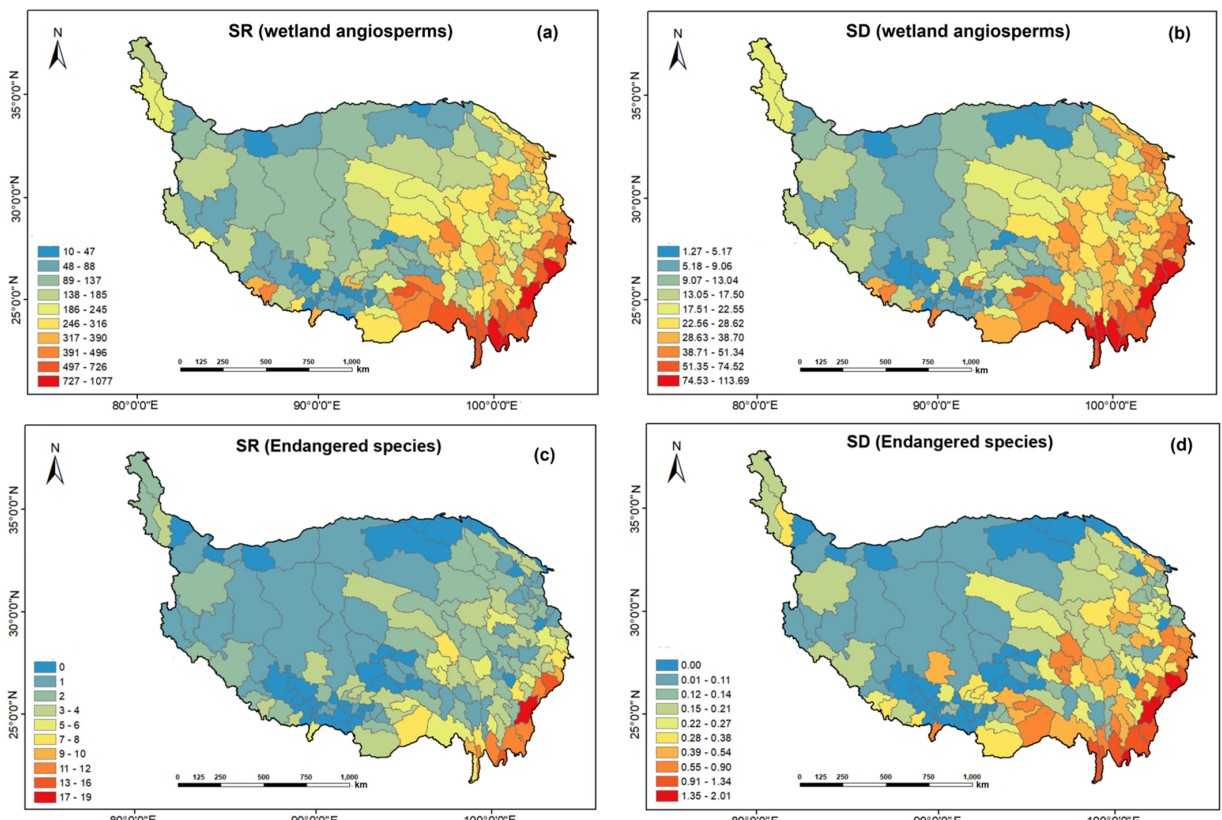

**Figure 3.** Distribution pattern of species richness (SR) and species density (SD) of wetland angiosperms (including endangered species) in the Qinghai-Tibet Plateau, projected in ArcGIS 10.2. (**a**,**b**) wetland angiosperms Species Richness (SR) and Species Density (SD); (**c**,**d**) Endangered wetland angiosperms Species Richness (SR) and Species Density (SD).

Endangered wetland angiosperms diversity hotspots on the Qinghai-Tibet Plateau are concentrated in the Hengduan Mountains and southeastern Tibet, with low or no distribution in other areas. Species density (SD) showed a more pronounced pattern of regional differences than species richness (SR) (Figure 3c,d), but the spatial variation pattern was consistent.

### 3.3. Elevation, Longitude, and Latitude Patterns of Species Richness

The SR of wetland angiosperms gradually decreases with the increase of elevation and latitude (Figure 4a,b), on the contrary, it increases with the increase of longitude (Figure 4c). At the same time, SR changes significantly with elevation and longitude gradients. In general, elevation contributed the most to the change of species richness (25%), while latitude contributed the least to the change of species richness (3%).

### 3.4. Relationship between Environmental Variables and Species Richness

The SR of wetland angiosperms was positively correlated with AMT, AP, PDM, and EVC, and negatively correlated with TS, PS, Ratio_WA, and WA (Table 1) (Figure S1). We discussed the related contributions of three environmental variables, energy, water, and habitat, to the variation of wetland angiosperms richness. The results showed that the three environmental variables jointly explained 43.58% of the variation. Among them, the contribution rate from large to small is water (42.65%), energy (28.49%), and habitat (14.27%). It shows that the distribution pattern of wetland angiosperms species richness is mainly affected by two environmental variables: energy and water, especially water environmental variables (Figure 5).

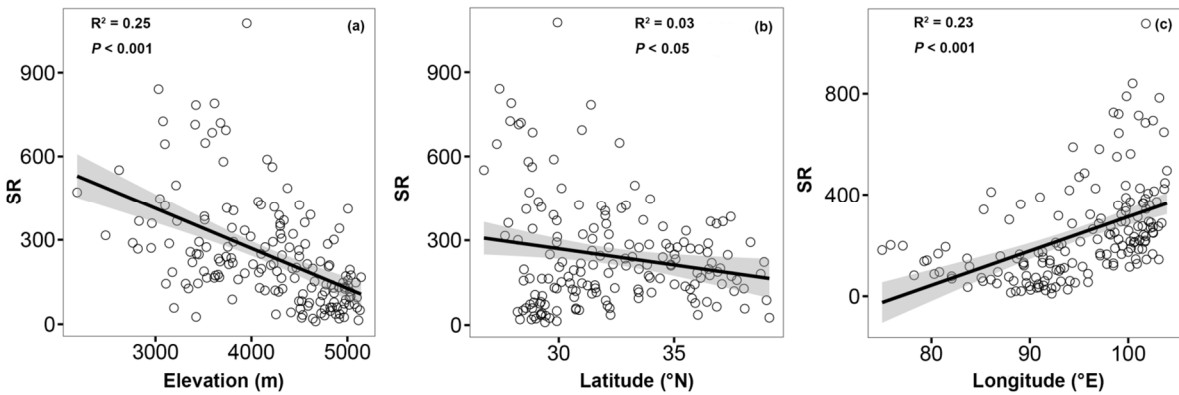

**Figure 4.** Elevation (**a**), latitude (**b**), and longitude (**c**) patterns of wetland angiosperms in the Qinghai-Tibet Plateau. LM model is used to fit the data, and the shaded areas represent the 95% confidence interval (N = 166).

**Table 1.** Correlation Analysis between species richness and three environmental variables: energy, water, and habitat.

| Environmental Variables | *p*-Value | $R^2$ |
|---|---|---|
| Energy | | |
| AMT | $p < 0.001$ | 0.29 |
| TS | $p < 0.001$ | 0.15 |
| Water | | |
| AP | $p < 0.001$ | 0.39 |
| PDM | $p < 0.001$ | 0.24 |
| PS | $p < 0.001$ | 0.20 |
| Habitat | | |
| Ratio_WA | $p < 0.05$ | 0.02 |
| EVC | $p < 0.001$ | 0.14 |
| WA | $p < 0.05$ | 0.02 |

Notes: AMT, annual mean temperature; TS, temperature seasonality; AP, annual precipitation; PDM, precipitation of driest month; PS, precipitation seasonality; Ratio_WA, the ratio of wetland area to county area; EVC, Elevation variation coefficient; WA, wetland area.

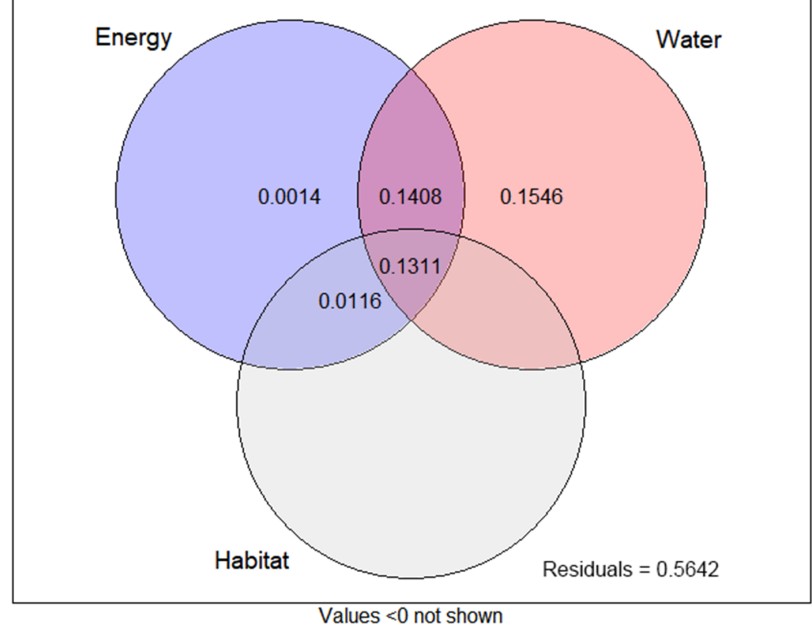

**Figure 5.** Variance partitioning (proportions) of species richness for Wetland angiosperms into the independent effects of energy, water, and habitat variables, and their overlaps.

## 4. Discussion

Based on field surveys, specimens data, online databases, and literatures, we comprehensively combed the checklist and county-level distribution pattern of wetland angiosperms and endangered species in the Qinghai-Tibet Plateau, analyzed the elevation, longitude, and latitude distribution pattern of wetland angiosperms richness, and explains its distribution pattern from three groups of environmental variables: energy, water, and habitat. The results showed that the diversity of wetland angiosperms was high, mainly hygrophytes, and the species richness decreased from southeast to northwest. Endangered wetland angiosperms are mainly distributed in the Hengduan Mountains and southeastern Tibet. Elevation and longitude had greater effects on wetland angiosperms diversity patterns, while latitudinal gradients had relatively weak effects. AMT and AP were the most important variables positively correlated with SR. Interestingly, SR shows a downward trend with the increase of Ratio_WA and WA. Energy and water contribute the most to the variation of species richness, while habitat contributes the least.

Asteraceae, Cyperaceae, Primulaceae, Poaceae, Gentianaceae, and Rosaceae account for a large number of species. The dominance of these families is consistent with the research results of different research geographical units (provinces, wetlands, and watersheds) on the Qinghai-Tibet Plateau [14,15,47]. The explanation is that these families are all plant macro-families widely distributed throughout the world, such as Asteraceae and Poaceae, spread all over the world from the tropics to the poles [48,49]. Many studies have shown that the Qinghai-Tibet Plateau is the origin, distribution, and differentiation center of many species. These families and genera are highly diverse in the Qinghai-Tibet Plateau, genera *Gentiana*, *Pedicularis*, *Ligularia*, *Primula*, etc. [50–52].

Environmental filtering considers that a certain type of habitat only contains species suitable for living in the environment, and the environment determines which species in the regional species pool can enter and remain in the environment [53]. Studies have shown that the phylogenetic structure of vascular plants in most areas of the Qinghai-Tibet Plateau is clustered, and environmental filtering plays an important role in the assembly of communities on the plateau [37]. In the long-term evolution process, only the species that adapt to the special cold and humid habitat of the Qinghai-Tibet Plateau wetland can survive. We analyzed the common habitats of dominant families and found that they all have the characteristics of adapting to the harsh environment of plateau wetlands. For example, Cyperaceae are widely distributed in temperate and cold regions, have a narrow ecological breadth, and tend to pair with Poaceae in wet conditions [48]. The number of species in the genera *Carex* and *Kobresia* was relatively large in the Cyperaceae, the genus *Carex* is the largest genus of wetland angiosperms species on plateau and one of the largest and most widely distributed in the world, Plants of the genus *Carex* are adapted to a wide variety of habitats and are a dominant population in habitats such as lakesides and marshes [54]. *Ranunculus* section *Batrachium* species are generally found in aquatic or semi-aquatic habitats [55]. *Batrachium bungei* communities are one of the most widespread community types on the Qinghai-Tibet Plateau. Except for Zayu, Medog, and the northern part of the Qiangtang Plateau, it is widely distributed in other regions. It generally grows in a variety of habitats such as lakes, rivers, ponds, meadow puddles, and stagnant depressions [48].

Because most orchid species are threatened or endangered in the wild, the relative scarcity of members of this family has given them special protection status [56]. This is also illustrated by the largest proportion of orchid species in the endangered wetland angiosperms in this study. In the survey of threatened orchids in China, it was found that habitat loss, degradation, and fragmentation were the main factors that threatened the vast majority of species [57]. Climate change is susceptible to threats to orchids because of their narrow range and specific symbiotic relationships with fungi and pollinators [58]. Climate change will further limit the availability of suitable habitats while increasing threats such as drought and weed spread [58]. The Qinghai-Tibet Plateau undoubtedly exacerbates this trend due to the uniqueness, primitiveness, and fragility of its ecosystem, as well as its high

sensitivity to global climate change. *Rhodiola* L. genus belongs to the Crassulaceae family and contains about 70 species, mainly distributed in the alpine regions of the northern hemisphere, especially the Qinghai-Tibet Plateau and adjacent high-altitude areas [59]. Different *Rhodiola* species have long been used in traditional medicine to treat various ailments. But in recent years, the commercialization of some species has led to an increase in the demand and development of *Rhodiola* plants [60]. At the same time, economic and construction activities such as grazing may also have a certain impact on the number of *Rhodiola* species [59,60]. This study also found that some species of Crassulaceae are narrowly distributed, and may be on the verge of extinction due to the intrinsic factors of the species itself.

Our investigation found that the distribution pattern of wetland angiosperms diversity follows the general climate law from warm and wet to cold and dry, from low to high average altitude, and from high to low ruggedness [61]. The abundance of wetland angiosperms in the Qinghai-Tibet Plateau decreases with the increase in elevation and latitude, following the general geographical distribution pattern [62,63]. In the investigation of the altitude gradient of aquatic plants on the Qinghai-Tibet Plateau, it was also found that aquatic plants decreased with the increase in altitude [30]. Elevation and latitude gradients are comprehensive variables, including environmental factors such as temperature, humidity, and light [30]. These composite variables also vary with altitude and latitude, for example, solar irradiance, temperature, and precipitation generally decrease with increasing latitude [30,64,65]. Some studies have pointed out that temperature affects plant diversity by affecting physiological responses of aquatic plants, such as seed germination and seasonal growth, and low temperature by freezing sediments, thereby limiting light penetration and gas exchange between space and water. The study showed a significant positive correlation between temperature and aquatic plants [30,66]. Our study also found that annual mean temperature (AMT) was an important environmental factor affecting wetland angiosperms diversity. In this study, wetland angiosperms species richness and longitude showed a significant positive correlation. The temperature of the Qinghai-Tibet Plateau gradually rises from west to East, and the precipitation gradually increases from the northwest to the southeast of the plateau [67]. With the increase of longitude, the wetland angiosperms plants show a significant increasing trend.

The distribution pattern of wetland angiosperms diversity in the Qinghai-Tibet Plateau is consistent with the previous research results. Species richness is higher in the East and south of the plateau, but lower in the northwest of the plateau [37,48,61,68]. A large number of studies have proved that climate is an important factor driving global and regional biodiversity patterns [4]. Hydrodynamics has been widely studied as a key driving factor for the spatial distribution of wetland vegetation [13]. As an important part of the wetland ecosystem, water plays an important role in wetland vegetation diversity [13]. The impact of climate on the wetland ecosystem is mainly reflected in rainfall and temperature, which affects the availability of water for plants. Our results show that energy and water can better explain the distribution pattern of wetland angiosperms richness. Previous studies have also shown that energy and water are the most important determinants of the distribution pattern of gymnosperms' richness in the Himalayan region [69]. We generally believe that areas rich in energy and water resources have high species diversity [69]. In this study, AMT and AP were used as the most important representative factors of energy and water. It was found that with the increase of AMT and AP, species richness increased significantly. There is an obvious spatial pattern of precipitation and temperature in the Qinghai-Tibet Plateau [48,70,71]. Based on the analysis of five alpine wetlands and climate change in the Qinghai Tibet Plateau Based on remote sensing data, it is concluded that AMT and AP are the key factors affecting the vegetation of alpine wetlands [72].

It is generally believed that there is a positive correlation between species diversity and area [30]. However, we were surprised to find that wetland angiosperms richness was less correlated with wetland area ($R^2 = 0.02$). The insufficient prediction of the relationship between wetland plant richness and wetland area may be due to the fact that most plants

grow on the lake bank, resulting in the species-area relationship is not very obvious [62]. Some studies have also proved that the relationship between wetland plant diversity and colonization area is greater than that of the total surface area of the wetland [62,73]. Heterogeneous habitats promote species coexistence and diversity by providing more niche space and more opportunities for reproductive isolation [23,74]. In this study, topographic heterogeneity is positively correlated with wetland angiosperms richness, which is consistent with the habitat heterogeneity hypothesis. However, we found that habitat variables explained only 14.27% of species richness. We need to consider the combined effects of habitat variables, temperature, and precipitation. Although higher terrain (larger elevation differences with shorter horizontal distances) may create certain geographical barriers to species dispersal and migration routes, thereby promoting species diversity, it needs to be combined with climatic factors [4,75]. For example, although the terrain in the northwestern part of the plateau is also complex, the climatic conditions are cold and dry, and the plant diversity is low [4]. At the same time, we need to consider another aspect. In this study, the variance explained the rate of habitat variables to species richness is very low, which may be because the three selected habitat variables cannot well represent the heterogeneity of wetland habitats. Studies have shown that the presence of water (seasonal and permanent) is a major determinant of wetland habitat and associated vegetation [76]. Seasonal water level fluctuations play a key role in promoting the diversity and distribution of aquatic and hygrophilous vegetation [77]. In the future, we will focus on the impact of seasonal hydrology on wetland angiosperms diversity in a plateau.

The endangered wetland angiosperms on the Qinghai-Tibet Plateau are mainly distributed in the Hengduan Mountains and southeastern Tibet, which is consistent with the hot spots of wetland angiosperms diversity. When determining priority protection areas, we should focus on the above areas. However, the identification of priority protected areas often takes into account various biological indicators. In addition to considering species richness and species density indicators, we will comprehensively consider species composition, function, endemicity, and threat level [78,79]. As the threats to endangered plants are often multifaceted, such as illegal logging, habitat loss, and fragmentation, climate change, etc., these threats may occur simultaneously, and protection measures need to be comprehensively considered and adapted to local conditions [58].

We attempted to compare the species composition and diversity patterns of the Qinghai-Tibet Plateau wetlands with other wetlands in the world. In terms of species composition of wetland angiosperms, it was found that Asteraceae, Cyperaceae, and Poaceae were three typical families of wetland angiosperms. For example, Asteraceae, Cyperaceae, and Poaceae account for 49% of the angiosperms in the southern Brazilian plateau marsh [80]. In the study of wetland plants in the Gasa District of Northern Bhutan in the eastern Himalayas, it was found that Asteraceae, Cyperaceae, and Poaceae were also dominant families [81]. In the survey of plant diversity in the Pantanal wetland, it was found that Asteraceae, Cyperaceae, and Poaceae were also the families with higher species richness [82]. Of course, different wetland ecosystems have unique species compositions due to their special geological history and modern environmental conditions. For example, orchids are the main family of plants in the Gasa District of Northern Bhutan in the eastern Himalayas [81]. On the contrary, among the angiosperms in the Qinghai-Tibet Plateau wetlands, orchids only account for 1.4%. We also try to find the key environmental variables that drive changes in wetland plant diversity. In this study, temperature and precipitation were the most critical factors influencing the changes in wetland angiosperms richness in the plateau, which was confirmed in the study of wetland plant diversity in South Africa. In the study of the changing laws of wetland plant diversity in South Africa, the Kingdoms of Lesotho, and ESwaitini, it was found that the more precipitation, the more water resources available and the more wetland habitat types [83]. Therefore, the larger the wetland environmental gradient, the increase of wetland plant ecological niche and the increase of wetland plant diversity. This suggests that there is a direct relationship between wetland species diversity and the climatic regime over large areas [83]. When

studying the effects of climate on species richness of freshwater plants in Europe and North America, it was found that annual mean temperature and annual precipitation showed the highest relative impact, for Europe (57.4% and 9.8%) and North America (51.6% and 31.4%), respectively [66].

## 5. Conclusions

This study systematically combed the checklist and distribution pattern of wetland angiosperms in the Qinghai-Tibet plateau for the first time, discussed the elevation, longitude and latitude gradients of wetland angiosperms diversity, and revealed the different effects of environmental variables on wetland angiosperms diversity. The results showed that the diversity of wetland angiosperms was high, showing a decreasing distribution pattern from southeast to northwest. The abundance of wetland angiosperms decreases with the increase of elevation and latitude, and increases with the increase of longitude. Energy and water can well explain the differences in the distribution pattern of wetland angiosperms. Endangered plants of Orchidaceae and Crassulaceae account for a relatively large proportion. When we designate the priority protection area for endangered wetland angiosperms, we recommend that we focus on the Hengduan Mountains and southeastern Tibet. In future research, we will focus on the impact of seasonal hydrological fluctuations on wetland plant diversity. At the same time, we will systematically study the phylogenetic diversity and functional diversity of wetland angiosperms, so as to provide experience and reference for the protection of alpine wetland ecosystems on the Qinghai-Tibet Plateau.

**Supplementary Materials:** The following supporting information can be downloaded at: https://www.mdpi.com/article/10.3390/d14100777/s1, Figure S1. Linear relationship between explanatory variables and wetland angiosperm species richness (SR).

**Author Contributions:** Data curation, Y.L. and Y.Z.; formal analysis, Y.L. and Y.Z.; funding acquisition, X.L. and Q.W.; investigation, Y.L., Y.Z., F.L., X.L. and Q.W.; methodology, Y.L. and Y.Z.; supervision, F.L., X.L. and Q.W.; writing—original draft, Y.L.; writing—review and editing, Y.L. and Y.Z. All authors have read and agreed to the published version of the manuscript.

**Funding:** This study was supported by the Second Tibetan Plateau Scientific Expedition and Research (STEP) program (2019QZKK0502), the National Natural Science Foundation of China (Grant Number 31860046), High-Level Talent Introduction Project of Tibet University (2020-1-20) and High-level graduate talent training program of Tibet University (2020-GSP-B016).

**Institutional Review Board Statement:** Not applicable.

**Informed Consent Statement:** Not applicable.

**Data Availability Statement:** Data (The Species checklist of wetland angiosperms in the Qinghai-Tibet Plateau) is available from the corresponding author upon request.

**Acknowledgments:** The author thanks all the students in the laboratory for their help in field sampling, Meiying Gong, Xiaoliang Jiang, Haoming Wu, Xuanfeng Wu and Leilei Luan for their help in data analysis.

**Conflicts of Interest:** The authors declare no conflict of interest.

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
