# Peer review of "Diversity Patterns of Wetland Angiosperms in the Qinghai-Tibet Plateau, China"

_diversity, doi:10.3390/d14100777_

Round 1

Reviewer 1 Report

This Q-T Plateau is a vast area, often "overshadowed" by research focus on the mountainous southern rim. Natural grassland systems (whether alpine, moorland or tropical temperate) are extremely valuable systems of water production, livelihoods, carbon sequestration, and biodiversity repository. High plateaux with low topographic diversity in particular are under-appreciated and often highly transformed and degraded compared to rugged mountain systems. Thus this paper held a lot of promise in terms of topic, but major changes are needed before it can be considered scientifically sound.

MAJOR COMMENTS:

1. The study area as presented is huge - so extremely huge and complex that it is very hard to understand what the rationale for its use in this study is. It is extremely hard to find some sort of testable hypothesis in such a complex landscape focusing on wetlands, which are complex azonal environments themselves. I really suggest that the study area should be partitioned into what is true Plateau (low topographic variability) and what is true Mountainous (high topographic variability) and exclude the latter from this paper (maybe as a second paper). The latest GMBA mountain database would help separate the two (based on topography). 

2. There is a lot of ambiguity as to what the study is trying to achieve. The water-energy-habitat idea is weakly presented, and the results don't really tell us anything new other than general global biogeographic patterns we know already. There is no novelty as relating to the Plateau. It would be more meaningful to consider diversity linked to seasonality, water availability, season inundation / permanent inundation, etc. - these are better predictors of diversity for wetlands. 

3. Given the geographic space the study area occupies, Fig. 3 is redundant within itself: the Elevation and Latitude graphs are saying the same thing, because Elevation increases with Latitude based on their study area definition so of course diversity will decrease, and likewise Elevation decreases with Longitude so of course diversity will increase going W. Better predictors would be different types of wetlands likely present on the Plateau (see classical wetland descriptions for examples, that could be used in the study perhaps), and the wetland functional traits / guilds associated with the different types of wetlands. But, just using the Plateau would give a better picture of trends across the landscape, but it should be pegged to something less nebulous in the interpretation. 

4. The Discussion makes a better Intro than the Intro, and much of it could be a better departing point than the current Intro.

5. There could be some comparisons with other vast high elevation, natural grassland-moorland systems with embedded azonal wetlands - such as the Prairies (N America), Highveld (South Africa), Pampas (S America), Central Asian Steppe (Ukraine, etc.), in terms of biodiversity patterns, trends, species compositions. This would add a lot of value to the paper. 

Summary: The authors should consider completely restructuring the paper to rather focus on (i) the Plateau only, (ii) plant diversity patterns on the Plateau linked to types of wetlands on the Plateau; (iii) some comparisons with similar systems elsewhere to highlight any novelties or commonalities.

OR the paper could be presented as just a "Checklist of Wetland Plant Taxa on the Q-T Plateau", and leave it at that (no need to try and test something perhaps untestable in such a complex and vast region). 

MINOR COMMENTS:

1. Some English editing needed in the Abstract and here and there in the body of the paper. Some careful proofing to pick up a few typos, and there is some "casual" language that could be better worded for a journal. Given the paper needs major restructuring I have not listed them all individually. 

2. The Abstract is weak and contradictory and needs to be reworked thoroughly once the paper is revised. 

Author Response

MAJOR COMMENTS:

Point 1: The study area as presented is huge - so extremely huge and complex that it is very hard to understand what the rationale for its use in this study is. It is extremely hard to find some sort of testable hypothesis in such a complex landscape focusing on wetlands, which are complex azonal environments themselves. I really suggest that the study area should be partitioned into what is true Plateau (low topographic variability) and what is true Mountainous (high topographic variability) and exclude the latter from this paper (maybe as a second paper). The latest GMBA mountain database would help separate the two (based on topography).

Response 1: I completely agree with the reviewer's comments. The Qinghai-Tibet Plateau's young geological history, unique alpine environment, complex wetland system and special plateau climate provide special ecological and geographical conditions for the growth of wetland plants. As you said, it's really hard to find a rationale in such a huge and complex area, and we've been working hard to find a way to solve it. Based on the construction of the Qinghai-Tibet Plateau wetland plant database, this paper comprehensively summarizes the angiosperm list and county-level distribution information of the Qinghai-Tibet Plateau wetland. We have done a lot of preliminary work, trying to use climate and habitat factors to explore the distribution pattern of wetland plant diversity at the large-scale of the Qinghai-Tibet Plateau. In the following research, we will divide the Qinghai-Tibet Plateau into real plateaus and real mountains according to your opinions, and try to explore the pattern of wetland plant diversity more accurately from two different regions.

Point 2: There is a lot of ambiguity as to what the study is trying to achieve. The water-energy-habitat idea is weakly presented, and the results don't really tell us anything new other than general global biogeographic patterns we know already. There is no novelty as relating to the Plateau. It would be more meaningful to consider diversity linked to seasonality, water availability, season inundation / permanent inundation, etc. - these are better predictors of diversity for wetlands.

Response 2: Thank you very much for the comments of the reviewer. Based on previous studies, this paper selects common environmental variables for preliminary analysis of wetland plant diversity. In the study of plant diversity and community composition of wetlands on the Qinghai-Tibet Plateau, it was found that latitude, Annual mean temperature and annual precipitation were the main predictors of plant taxonomic and phylogenetic diversity (Cui et al., 2021). In the study of the altitude pattern of aquatic plants on the Qinghai-Tibet Plateau, it was found that the annual average temperature was the most important variable affecting the taxonomy, phylogeny and functional diversity of species (Zhou et al., 2021). In a study of aquatic plant community composition and species richness in 454 lakes in Minnesota, it was found that macrophyte community composition and species richness were mainly affected by regional (climate) and local patterns (water quality) (Alahuhta ,2015). The three types of prediction indicators proposed by the reviewer are very targeted and meaningful. In the following research, we will use the indicators you proposed to analyze the diversity pattern (At present, the area of the county-level units we use is different, and we will build a grid to eliminate the impact of regional inconsistencies, which may be more scientific and reasonable).

Point 3: Given the geographic space the study area occupies, Fig. 3 is redundant within itself: the Elevation and Latitude graphs are saying the same thing, because Elevation increases with Latitude based on their study area definition so of course diversity will decrease, and likewise Elevation decreases with Longitude so of course diversity will increase going W. Better predictors would be different types of wetlands likely present on the Plateau (see classical wetland descriptions for examples, that could be used in the study perhaps), and the wetland functional traits / guilds associated with the different types of wetlands. But, just using the Plateau would give a better picture of trends across the landscape, but it should be pegged to something less nebulous in the interpretation.

Response 3: The point of view put forward by the reviewer is indeed very reasonable. It is common for species diversity to decrease with altitude and latitude. We found in the literature that the species richness of alpine grasslands on the Qinghai-Tibet Plateau increased with latitude and longitude (Yang, 2004), while the plant species diversity of Aerjin Mountain Nature Reserve did not change significantly along latitude, longitude and latitude (Dong et al., 2019). Due to the peculiarity of the Qinghai-Tibet Plateau and the complexity of wetland types, we would like to try to explore the changes in the lower-level diversity pattern from three gradients.

At the same time, according to the comments of the reviewers, we discussed the impact of different wetland types on the diversity pattern. We mainly discuss two types of wetlands: riverine-lacustrine (riverine and lacustrine are grouped together) and swamps, using the wetland area ratio as a predictor, with county-level units. Among them, Ratio_SW is the ratio of swamp wetlands to the total wetland area, and Ratio_RL is the ratio of riverine-lacustrine type wetlands to the total wetland area. We found a very low correlation between species richness and wetland area ratio (R2=0.01675). In the study on the relationship between vegetation characteristics and environment of six wetland types on the Qinghai-Tibet Plateau, it was found that the species richness of wetlands has a strong correlation with the location of wetlands, while there is no significant difference in species richness among the six wetland types (Li et al., 2016).

Point 4: The Discussion makes a better Intro than the Intro, and much of it could be a better departing point than the current Intro.

Response 4: Thank you for the reminder of the reviewer, we have revised the discussion part, hoping to meet your requirements.

Point 5: There could be some comparisons with other vast high elevation, natural grassland-moorland systems with embedded azonal wetlands - such as the Prairies (N America), Highveld (South Africa), Pampas (S America), Central Asian Steppe (Ukraine, etc.), in terms of biodiversity patterns, trends, species compositions. This would add a lot of value to the paper.

Response 5: Thank you very much for the comments of the reviewers, which brought us a lot of good ideas. We plan to carry out this research as a separate paper. This is a very meaningful work, thank you!

MINOR COMMENTS:

Point 1: Some English editing needed in the Abstract and here and there in the body of the paper. Some careful proofing to pick up a few typos, and there is some "casual" language that could be better worded for a journal. Given the paper needs major restructuring I have not listed them all individually.

Response 1: Thank you for your comments, we have made corresponding changes, we hope to meet your requirements, thank you!

Point 2: The Abstract is weak and contradictory and needs to be reworked thoroughly once the paper is revised.

Response 2: Thank you very much for the comments of the reviewer. The abstract mentions that wetland plant diversity is not related to wetland area, and in the first manuscript, the analysis mentioned that species richness is negatively correlated with area, so we have made changes.

Reviewer 2 Report

Manuscript ID: diversity-1870182

Title: Diversity patterns of wetland angiosperms in the Qinghai-Tibet Plateau, China.

It is not totally clear whether the manuscript in present form represent an original article or Review, since all of the data about plant species come from the records of other researchers. I suggest the authors to clear this issue. For the Review, the list of references is too short and does not cover other areas, while for the original article the authors should better define in the Introduction and Methods what are the results of previous researches and what are their contributions. However, the topic of the manuscript is important, since it is a study on diversity in wetlands. The climate changes and increasing demands for water represent a threat to the wetlands.

The manuscript deals with influence of available environmental variables on the distribution and diversity of plants. For these reasons, the manuscript fits well into the scope of the journal as well as the Special Issue. I suggest major changes, before the manuscript is processed further.

I recommend inclusion and discussion of the following questions into your work:

How is the plant diversity related and distributed in different types of the wetlands? You mention wet meadows, swamp, river wetland and lake wetlands.

What kind of management or protection is needed to maintain this diversity?

Rather than listing the largest families and genera I would expect focusing on Red List plant species. Information of the presence and distribution of rare, endangered or vulnerable wetland species would be much more valuable and useful for the conservation of the specific wetlands types as well as suitable management practices. This would also enable to write more suitable conclusions.

Some results from the Supplementary material could be probably moved to the main text since the manuscript is considerably short. The Supplementary material is hard to open.

Specific Comments:

Materials and Methods:

How were plant species classified to categories? Please add reference or describe the method in detail.

Results:

Table 1.

Data about the largest families and genera do not have much practical value for nature conservation nor diversity. Rather than listing the largest families and genera I would expect focusing on Red List plant species of China or narrower area.

Information of the presence and distribution of rare, endangered or vulnerable wetland species would be much more valuable and useful for the conservation of the specific wetlands types as well as suitable management practices.

Figure 2.

I suggest transforming the present data and relate them according to the surface area of specific counties.

Ln 293: genus Ranunculaceae - please correct

Conclusion:  

Ln 364: In which way this work provides experience and guidance for the protection? Please explain.

What do you suggest?

References:

I suggest inclusion of additional references, especially those from other parts of the world than China. Almost two thirds of the references are written exclusively by Chinese researchers, among them several in Chinese. Inclusion of other researches would better connect the article with international scientific audience.  

Author Response

Point 1: It is not totally clear whether the manuscript in present form represent an original article or Review, since all of the data about plant species come from the records of other researchers. I suggest the authors to clear this issue. For the Review, the list of references is too short and does not cover other areas, while for the original article the authors should better define in the Introduction and Methods what are the results of previous researches and what are their contributions.

Response 1: Thank you very much for the comments of the reviewer. Based on a large number of monographs, literature and online databases and field sampling data, this paper comprehensively sorts out the species list and county-level distribution information of angiosperms in the Qinghai-Tibet Plateau wetlands. Thanks to the scholars who have done a lot of work on the survey and arrangement of plants on the Qinghai-Tibet Plateau in the past 60 years, this paper is an original article, and we have supplemented the data source and analysis and arrangement process in the Methods section. In the first manuscript, this part of the content was placed in the supplementary material. We are very grateful for the comments of the reviewer, and we have put it back into the main text.

Point 2:  How is the plant diversity related and distributed in different types of the wetlands? You mention wet meadows, swamp, river wetland and lake wetlands. What kind of management or protection is needed to maintain this diversity?

Response 2: According to the reviewer's opinion, we discuss the impact of different wetland types on diversity patterns. We mainly discuss two types of wetlands: riverine-lacustrine (riverine and lacustrine are grouped together) and swamps, using the wetland area ratio as a predictor, with county-level units. Among them, Ratio_SW is the ratio of swamp wetlands to the total wetland area, and Ratio_RL is the ratio of riverine-lacustrine type wetlands to the total wetland area. We found a very low correlation between species richness and wetland area ratio (R2=0.01675). In the study on the relationship between vegetation characteristics and environment of six wetland types on the Qinghai-Tibet Plateau, it was found that the species richness of wetlands has a strong correlation with the location of wetlands, while there is no significant difference in species richness among the six wetland types (Li et al., 2016).

Because the species distribution data is based on county-level units, species checklist for specific wetlands are not included. Therefore, this paper does not discuss the relationship between different types of wetlands, and temporarily cannot give specific protection suggestions for different types of wetlands. In the following research, we will select representative wetlands of different wetland types on the Qinghai-Tibet Plateau, and try to find the key factors affecting the diversity of different types of wetlands by comparing and analyzing the species composition and diversity patterns of different types of wetlands.

Point 3:  Rather than listing the largest families and genera I would expect focusing on Red List plant species. Information of the presence and distribution of rare, endangered or vulnerable wetland species would be much more valuable and useful for the conservation of the specific wetlands types as well as suitable management practices. This would also enable to write more suitable conclusions.

Response 3: Thank you for your comments. We need to pay more attention to the status of endangered wetland plants, which is even more important for the conservation of wetland plant diversity. According to the Chinese Biodiversity Redlist of Higher Plants, we compiled a list of endangered wetland angiosperms on the Qinghai-Tibet Plateau, and drew a map of the distribution of endangered wetland angiosperms.

Point 4: Some results from the Supplementary material could be probably moved to the main text since the manuscript is considerably short. The Supplementary material is hard to open.

Response 4: Thank you very much for the comments of the reviewer. We have supplemented the relevant data according to the requirements of the reviewer and put it in the text, please check it.

Point 5: How were plant species classified to categories? Please add reference or describe the method in detail.

Response 5: We have put the relevant references and analysis steps in the Methods section, please check it.

Point 6: Data about the largest families and genera do not have much practical value for nature conservation nor diversity. Rather than listing the largest families and genera I would expect focusing on Red List plant species of China or narrower area.

Information of the presence and distribution of rare, endangered or vulnerable wetland species would be much more valuable and useful for the conservation of the specific wetlands types as well as suitable management practices.

Response 6: The comments of the reviewer are very helpful to us. We supplemented the information on the family, genera and distribution of endangered wetland plants as required, and made basic analysis.

Point 7: Figure 2. I suggest transforming the present data and relate them according to the surface area of specific counties.

Response 7: According to the comments of the reviewer, we have supplemented the distribution information of species density on the basis of species richness, please check it.

Point 8: Ln 293: genus Ranunculaceae - please correct

Response 8: Thank you for the reminder of the reviewer, we have made changes, please check it.

Point 9: Conclusion:  Ln 364: In which way this work provides experience and guidance for the protection? Please explain. What do you suggest?

Response 9: According to the comments of the reviewers, we have supplemented the relevant information on endangered wetland plants. We found that the Hengduan Mountains and southeastern Tibet are places with more endangered plants, and we should focus on them when formulating priority protected areas. At the same time, this paper comprehensively summarizes the angiosperms in the Qinghai-Tibet Plateau wetlands, which can provide basic data for the protection of wetland angiosperms.

Point 10: References: I suggest inclusion of additional references, especially those from other parts of the world than China. Almost two thirds of the references are written exclusively by Chinese researchers, among them several in Chinese. Inclusion of other researches would better connect the article with international scientific audience.

Response 10: Thank you for the comments of the reviewer, we have made supplementary references, please check it.

Round 2

Reviewer 2 Report

The authors have considered the comments and have improved the manuscript. The major part of the suggestions have been implemented, while some things that could not be solved at this stage have been explained.

I suggest to provide the Supplementary files that could be easily accessed to the readers and english-check by native speaker.

Author Response

Dear Reviewer:

Thank you for your comments on our manuscript entitled "Diversity patterns of wetland angiosperms in the Qinghai-Tibet Plateau, China" (ID: diversity-1870182). These comments are valuable for the revision and improvement of our manuscript, and have important guiding significance for our research.

Thanks to the reviewers for their professional opinions, there are some problems that cannot be solved temporarily in this paper, such as the comparison between different types of wetlands, we will deal with these problems in future work, so that our research will be more complete and valuable.

According to your comments, we will put some pictures of data processing on it for your convenience.

Thanks to the reviewers for their diligent work and hope that the revisions are approved.

Thanks again for your comments and suggestions.